# Protective Gear Negatively Impacts Police Officer Mobility, Stability, and Power Generation

**DOI:** 10.3390/jfmk10030344

**Published:** 2025-09-09

**Authors:** Katherine A. Frick, Philip J. Agostinelli, Frances K. Neal, Nicholas C. Bordonie, C. Brooks Mobley, JoEllen M. Sefton

**Affiliations:** 1Warrior Research Center, Auburn University, 301 Wire Rd., Auburn, AL 36832, USA; kaf0067@auburn.edu (K.A.F.); pja0007@auburn.edu (P.J.A.); fkn0001@auburn.edu (F.K.N.); ncb0039@auburn.edu (N.C.B.); 2Nutrabolt Applied and Molecular Physiology Laboratory, 301 Wire Rd., Auburn, AL 36832, USA; moblecb@auburn.edu

**Keywords:** occupational performance, law enforcement, safety gear

## Abstract

**Background:** Protective gear is a critical part of the police officer uniform. The required protective gear weighs over 9 kg and is rigid and bulky, creating deficits in physical performance essential for completing officer’s daily tasks and increasing risk of injury. Understanding the impedance the protective gear causes and how physical factors such as body composition increase this effect is critical to the safety and survival of the police officer. The purpose of this study was to evaluate the impact of protective gear on officer capabilities. **Methods:** Officers completed an 11-point assessment in two conditions: athletic attire (No Gear) and uniform + protective equipment (Gear). **Results:** Differences in power output (*p* < 0.001; *p* = 0.118), balance (*p* < 0.001; *p* = 0.771), functional movement (*p* = 0.002; *p* = 0.018), and flexibility (*p* < 0.001) were found between the two conditions. **Conclusions:** Decreased on-duty performance can affect officer safety and success. These results indicate the need for continued improvement of police officer safety equipment to ensure mobility and safety.

## 1. Introduction

Mandatory protective gear is essential for police officer work and safety [1,2]. Traditional gear includes a Kevlar bullet-resistant vest housing a body camera and radio and a duty belt with an extendable baton, pepper spray, handcuffs, a personal protection kit, a flashlight, a taser, a firearm, and additional ammunition clips (Figure 1, Table 1) [1,2]. Police officers often criticize the discomfort and cumbersome nature of the mandatory protective gear [1,3]. Research has established a link between the heavy protective vests and duty belts with lower back pain and decreased physical capabilities [1,3,4,5,6,7]. Studies indicate that 43% of Swedish active-duty police officers experience lower back pain one or more days per week [8]. Investigators identified the duty belt, body armor, weapon, and radio as the primary sources of discomfort [9]. Some regions in the United States have transitioned to tactical load-bearing vests as an alternative to conventional safety gear, but others have adhered to traditional equipment usually for reasons of cost and appearance [7,10,11,12]. Police departments that are more traditional in nature focus on the approachability of officers and tend to shy away from the use of tactical vests because of the militaristic look [10,13].

Research has investigated the impact of external load on physical capabilities on tactical populations such as firefighters [14] and the military [2,15]. Military-centric research indicates that sustained carriage diminishes overall performance and elevates physiological demands [16,17,18]. Research on traditional sports athletes suggests that weighted vests comprising as little as 7–8% of an athlete’s body weight affect running performance, stability, and mobility [19,20]. Few studies have specifically assessed the effect of protective vests and duty belts on the overall mobility and stability of police officers [10,13]. Previous research focusing on the weight of each protective component of police officers’ protective equipment suggests that the protective gear worn by U.S. police officers represents approximately 8–12% of their body mass [11]. This additional weight poses risks and declines in movement, speed, stability, and power output [11,21].

Efficient and effective movement, mobility, and strength output are important for completing police officers’ daily duty requirements [11]. Previous studies have demonstrated that protective gear diminishes physical capabilities, including stability, balance, movement recovery, gait, and power output [22,23]. These performance decrements are likely a result of the added weight, distribution and rigidity of protective vests and equipment which may impede an individual’s ability to move efficiently and correct balance loss [22,23]. Research has yet to evaluate the effects of the protective gear and extra weight on officers’ ability to move and complete their duties. Moreover, no research to date has evaluated how physical factors such as body composition interact with the required equipment to impede performance. Restricted physical capabilities due to police officer equipment influence the safety and survival of officers and the communities they serve.

Therefore, the primary objective of this study was to (1) assess the mobility, stability, and power output of police officers with and without protective gear; (2) determine which specific movements were affected; and (3) ascertain if body composition influenced the outcome of mobility, stability, and power output with and without police protective gear. We hypothesized that police officer safety gear and equipment, along with increased body fat percentage, would decrease stability, mobility, and power production.

## 2. Materials and Methods

Members of a local police department (*N* = 72/165; 43.6% of police department population) (4 females, 68 males, age of 36.9 ± 9.7 years, height of 178.6 ± 7.5 cm, and body mass of 95.1 ± 11.8 kg) volunteered for this study. The demographics of the 72 police officers volunteering for this study are detailed in Table 2. All participants read and signed the informed consent approved by an institutional review board. Participants completed a 30 min data collection session including body composition measurements, pain provocation during functional movement patterns, functional movements, vertical jump, and sit-and-reach testing. All assessments (except body composition measurements which were completed only once) were completed in athletic clothing (deemed “No Gear”) and completed again with uniform and protective gear (deemed “Gear”) (Table 1). Police officers were weighed in No Gear (95.1 ± 16.3 kg) and then Gear (104.6 ± 16.8 kg) to evaluate the weight of the gear alone (9.7 ± 1.4 kg). This project was a part of a larger overall fitness evaluation, previously detailed methods are summarized below [24].

### 2.1. Body Composition

Body composition measures were assessed by first measuring the height (cm) and mass (kg) of each participant using a SECA scale and stadiometer (SECA, Hamburg, Germany), followed by scanning using a pre-programmed whole-body scan of Dual-Energy X-ray absorptiometry (DEXA, Lunar iDXA, GE Healthcare, Chicago, IL, USA). DEXA was used to report body fat percentage (BF%). The body mass index (BMI) was calculated for each participant using the standard height and weight calculation of weight in kilograms divided by height in meters squared [25].

### 2.2. Balance Variables

The velocity of the path length (path length/time) balance variable was assessed using the Leonardo Mechanography platform (Leonardo Mechanograph, Novotec Medical GmbH, Pforzheim, Germany). A familiarization period was conducted in which the participant stood as still as possible on both legs (tandem) for 30 s on the force plates. Single-leg balance testing was conducted for 30 s on both the right and left legs with the eyes open for each assessment in No Gear and Gear conditions.

### 2.3. Lower Body Power Output

Lower body power output was assessed by completing two vertical jumps of maximum height on the Leonardo Mechanography platform (Leonardo Mechanograph, Novotec Medical GmbH, Pforzheim, Germany). The Leonardo Mechanography platform is a dual, side-by-side force plate instrument allowing for independent measures of right and left legs during balance, jumping, and additional movement patterns. Each forceplate (right and left) measures the independent forces projected through each leg by the corresponding force plate and returns the value for total power output and individual leg power output during each jump trial. Participants were instructed to stand as still as possible on the platform with each foot on the corresponding force plate until the researcher gave a cue; four jumps in total were completed, two each in the No Gear and Gear conditions.

### 2.4. FMS Pain Provocation Tests and Clearance Tests

Participants completed three movement patterns (FMS pain provocation tests) that suggested predictive risks to the shoulders and lower back to determine the possibility of heightened risk and identify potential kinetic dysfunctions: shoulder clearing for active scapular stability, spinal extension with a prone press-up, and spinal flexion with quadruped posterior rock. Each test was scored on a scale of zero or one, with zero indicating no pain experienced during the movement and one indicating pain present [26].

### 2.5. FMS Trunk Stability and Full-Body Kinetic Function

Trunk stability was assessed using the FMS modified pushup, while full-body kinetic function was evaluated using overhead squat. Scoring for these tests followed the FMS ordinal scoring method: a score of three indicated completion without compensation or deviation, two denoted mild compensation or inadequacy, and one indicated inability to complete the movement or significant compensation. Pain Provocation and Functional Movement Screen assessments were graded according to the FMS scoring system [26]. Researchers adhered to the FMS script and scoring system, agreeing on grading after familiarizing themselves with the protocol.

### 2.6. Lower Back and Hamstring Flexibility

Lower back and hamstring flexibility were evaluated using the sit-and-reach assessment [25]. Participants sat on the ground, feet against the base of the box, and pushed the bracket as far as possible with palms down, holding the stretched position for two seconds. Three recorded trials were conducted, with the length moved recorded in centimeters, and the No Gear and Gear conditions were compared.

### 2.7. Statistical Analysis

Assessments were statistically evaluated and the No Gear and Gear conditions were compared. Normality was assessed using Shapiro–Wilk tests and visualized through Q-Q plots. Taken together, these statistical assessments indicated that the data was generally normally distributed. Significance in this investigation was set a priori at a *p*-value ≤ 0.05 for all measures unless a Bonferonni adjustment was applied upon findings of significance.

### 2.8. Statistical Analysis—Gear vs. No Gear Conditions

A linear regression model was used to examine the effect of Gear vs. No Gear on the balance measure of the velocity of path length. A repeated-measures MANOVA was used to evaluate lower body power output (vertical jump height, vertical jump power, and power output per kilogram). Independent bilateral leg power output was assessed in a repeated-measures Analysis of Variance (ANOVA). A McNemar’s test was conducted to assess the occurrence of pain through the three FMS pain provocation tests. A Wilcoxon Signed-Rank Test was conducted to assess both truck stability (FMS modified pushup) and full-body kinetic function (FMS overhead squat). Lower back and hamstring flexibility by way of the sit-and-reach test was analyzed using a dependent *t*-test.

### 2.9. Statistical Analysis—Body Fat Percentage and BMI Interactions

To evaluate the interaction of BMI and body fat percentage and balance variables (standard ellipse area, path length, and velocity of path length), a repeated-measures Multivariate Analysis of Covariance (MANCOVA) was conducted. Additionally, a MANCOVA was used to evaluate the interaction of BMI and body fat percentage and lower body power output of the vertical jumps. An additional repeated-measures Analysis of Covariance (ANCOVA) was conducted to evaluate the interaction of BMI and body fat percentage and independent bilateral leg power output. Logistic regressions were completed to assess body fat percentage and BMI during the FMS pain provocation tests. Ordinal regressions were completed for the assessment of movement FMS patterns (FMS modified pushup and FMS overhead squat). To assess BMI and body fat percentage’s interaction with lower back and hamstring flexibility (sit-and-reach), a repeated-measures ANCOVA was conducted.

All statistical analyses were conducted in R Statistical Program Software Version (4.2.2 (RStudio; Boston, MA, USA)) using the psych 2.3.3, lattice 0.20.45, ggplot2 3.4.4, dplyr 1.1.2, stats 4.2.2, and tidyr 1.3.0 packages.

## 3. Results

### 3.1. Balance

The path length mean velocity (Vmean) was assessed as the static balance dependent variable with (Gear) and without (No Gear) required protective gear. Dominant single-leg stance was used for this assessment. Data was also separated into measures comparing self-reported dominant leg results for additional examination (Table 3). The linear regression model was not significant (F_(1,142)_ = 2.05; *p* = 0.15), with the condition only accounting for 1.4% of the variance in the path length mean velocity (R^2^ = 0.014). On average, participants in the Gear condition had lower values than those in the No Gear condition (B = −0.60; SE = 0.42), but this difference was not statistically significant (t_(142)_ = −1.43; *p* = 0.15).

### 3.2. Vertical Jump

Static vertical jump was assessed to investigate the dependent variables of jump height, vertical jump power output per kilogram, and independent bilateral leg power output in the No Gear and Gear conditions. A repeated-measures MANOVA revealed a significant multivariate effect of gear (Pillai’s Trace = 0.586; F_(1;71)_ = 100.35; *p* < 0.001), indicating that performance differed between the Gear and No Gear conditions across the vertical jump outcomes. A significant gear × jump measure interaction (Pillai’s Trace = 0.730; F_(2;70)_ = 94.78; *p* < 0.001) suggested the effect of gear varied across jump height, jump power, and power/kg. Univariate ANOVAs confirmed significant main effects of Gear (F_(1,71)_ = 100.35; *p* < 0.001) and measure (F_(2,142)_ = 1363.98; *p* < 0.001), as well as a significant gear × measure interaction (F_(2,142)_ = 35.33; *p* < 0.001). Greenhouse–Geisser corrections indicated that these effects remained robust despite violations of sphericity.

Police officers jumped higher and generated significantly more power in the No Gear condition (jump height: *t*_71_ = 13.51; *p* < 0.001 (Figure 2A); power output per kg: *t*_71_ = 7.13; *p* < 0.001 (Figure 2B); right leg power output: *t*_71_ = −5.04; *p* < 0.001). No difference was found between the Gear and No Gear conditions when comparing the power output of the left leg (*t*_71_ = −1.63; *p* = 0.11). There was no significant difference in the non-dominant legs (*t*_71_ = −1.58; *p* = 0.118) when comparing dominant to non-dominant legs in the static vertical jump, but there was a significant difference when comparing the No Gear and Gear conditions (*t*_71_ = −5.28; *p* < 0.001) in the dominate leg. There was a decrease in power output during the static vertical jump in the Gear condition compared to No Gear.

### 3.3. FMS Movements

Assessment of joint pain of the shoulders, hips, and lower back, stability, and basic movement revealed no statistical significance between conditions for the posterior rock (*X*^2^ = 0; *df* = 1; *p* = 1), supine press up (*X*^2^ = 0.57; *df* = 1; *p* = 0.45), or right (*X*^2^ = 0.0; *df* = 1; *p* = 0.34) or left (*X*^2^ = 0.17; *df* = 1; *p* = 0.68) shoulder clearing tests. Functional movement was assessed using the modified pushup and the overhead squat in the Gear and No Gear conditions. The modified pushup (*V* = 161.5; *p* = 0.003) and overhead squat (*V* = 256.5; *p* = 0.019) were both statistically different between conditions, with higher movement proficiency and efficacy in the No Gear condition. Differences in the mobility assessments between the two conditions can be seen in Figure 3A,B and Table 4. Analysis of the sit-and-reach assessment was also significantly different between the two trials (t_71_ = 12.41; *p* < 0.001; mean difference =7.08; 95% CI [5.95, 8.22]; *d* = 1.46; r = 0.83), with the police officers in the No Gear condition being able to reach farther than those in the Gear condition (Figure 3C).

### 3.4. Effect of Body Fat Percentage and BMI on Assessments

Body mass was significantly different between the No Gear and Gear conditions (*t*_71_ = −54.31; *p* < 0.001) with officers weighing 9 kg/20 lbs more when wearing the protective gear. Pearson’s r correlation showed a significant interaction between body fat percentage and BMI (r(70) = 0.54; *p* < 0.001). Body fat percentage and the calculated body mass index were also evaluated in the assessments of both the No Gear and Gear conditions.

An ordinal logistic regression was conducted to examine the association of gear (Gear vs. No Gear), body fat percentage, and task (overhead squat vs. modified pushup) on performance scores, including all two-way and three-way interactions. The model included 288 observations. The results indicated that higher body fat percentage was associated with lower odds of achieving a higher score (OR, 0.86; 95% CI, 0.78–0.94; *p* < 0.001). Task type significantly influenced performance, with FMS overhead squat yielding lower scores compared with FMS modified pushup (OR, 0.015; 95% CI, 0.001–0.17; *p* = 0.006). Gear condition alone was not significantly associated with performance (OR, 0.78; 95% CI, 0.10–6.06; *p* = 0.88). Body fat percentage × task interaction was significant (OR, 1.12; 95% CI, 1.01–1.25; *p* = 0.038), suggesting that the negative effect of body fat percentage was stronger for the pushup than for the overhead squat. No other two-way and three-way interactions were significant (all *p* > 0.05). Threshold estimates for the ordinal categories were −7.16 for 1/2 and −3.89 for 2/3. These results indicate individuals with higher body fat percentages had lower FMS overhead squat scores. The interaction between body fat percentage and task suggests that body composition may have impacted performance.

A second ordinal logistic regression was conducted to examine the association of gear condition (Gear vs. No Gear), the body mass index (BMI), and task (overhead squat vs. modified pushup) on performance scores, including all two-way and three-way interactions (n = 288). These results indicated that no variables, BMI (OR, 1.00; 95% CI, 0.91–1.10; *p* = 0.93), gear condition (OR, 6.17; 95% CI, 0.21–183.28; *p* = 0.38), or task type (OR, 1.27; 95% CI, 0.02–83.53; *p* = 0.91), were significantly associated with performance scores. All interaction terms, including BMI × task (OR, 0.95; 95% CI, 0.83–1.11), gear × task (OR, 0.51; 95% CI, 0.01–103.61), gear × BMI (OR, 0.96; 95% CI, 0.83–1.12), and the three-way interaction (OR, 1.03; 95% CI, 0.85–1.24), were also nonsignificant (all *p* > 0.05). Threshold estimates for the ordinal categories were −2.57 for 1/2 and 0.55 for 2/3. BMI, gear condition, and task type did not significantly influence performance scores in the current study, and there was no evidence of interactions between these factors. The wide confidence intervals indicate considerable uncertainty in estimating these effects.

A repeated-measures MANCOVA was conducted to examine the effect of the Gear and No Gear conditions on vertical jump performance outcomes (jump height, vertical jump power, and vertical jump power relative to body mass (kg)), while controlling for the body mass index (BMI) and body fat percentage. The multivariate test revealed significant main effects of condition, BMI, and BF% on the combined dependent variables (Wilks’ Λ = 0.85, F_(3,136)_ = 8.28, *p* < 0.001; Wilks’ Λ = 0.36, F_(3,136)_ = 79.62, *p* < 0.001; and Wilks’ Λ = 0.61, F_(3,136)_ = 29.17, *p* < 0.001, respectively). Follow-up univariate ANCOVAs indicated that condition significantly influenced jump height (F_(1,138)_ = 12.41; *p* < 0.001; partial η^2^ = 0.72), with participants demonstrating lower jump height in the Gear condition compared to the No Gear condition. Condition did not significantly affect jump power (F_(1,138)_ = 0.40; *p* = 0.53; partial η^2^ = 0.48) or relative jump power (per kg body mass) (F_(1,138)_ = 1.71; *_p_* = 0.19; partial η^2^ = 0.42). A higher BMI and body fat percentage were consistently associated with poorer vertical jump performance across models (Figure 4). Body fat percentage demonstrated large effects on jump height (partial η^2^ = 0.40), absolute power (0.29), and relative power (0.34) (Figure 4). BMI contributed smaller but significant effects, particularly for absolute power (partial η^2^ = 0.32) (Figure 4). These results indicate that wearing gear significantly reduced jump height but did not significantly affect vertical jump power or power relative to body mass when controlling for BMI and BF%. Body fat percentage emerged as a strong negative predictor of vertical jump performance across outcomes, whereas BMI showed smaller, but still meaningful, associations.

## 4. Discussion

This research aimed to explore the impact of traditional police protective gear on the mobility, stability, and power output of law enforcement officers. Our results revealed diminished officer physical performance when outfitted with traditional mandatory protective equipment. Significant decreases across all assessments were found when comparing the No Gear condition to the Gear condition. Officers wore their own protective gear during this study to assure that gear was appropriately sized and fitted for each individual (Figure 1) and to ensure applicability of our findings. Our study highlights the substantial negative influence of protective equipment on officers’ physical capabilities, operational effectiveness, and overall safety.

Assessing mobility limitations caused by acute and prolonged use of heavy protective gear is important in the overall health and ability of a police officer to adequately complete their policing tasks. It is important to understand the implications of these decreased functional movements and the role they may play in increased risk of developing a musculoskeletal injury. Research into the effects of load carriage and heavy protective gear indicates reduced mobility in full body function and also in key joints such as the shoulders, torso rotation, low back, and hips [7,10,11,27]. The previous literature also indicates that long-term wearing of heavy protective gear can decrease overall mobility and increase health concerns related to mobility and movement and lead to both acute and chronic lower back pain [11]. Low FMS scores have been shown to indicate higher injury risk, particularly in highly performing and tactical athletes [28,29,30]. Both functional assessments (FMS) conducted in the current study revealed an overall lowered ability to perform the tasks while in gear. Nearly 27% of the sample population had decreased trunk stability and functionality during the modified pushup task when wearing protective gear (Table 4 and Figure 2A). This indicates a decrease in the ability to maintain trunk stability throughout the movement pattern, which can correspond to a decreased ability to execute essential law enforcement tasks such as foot pursuits, crowd control, and subject apprehension while in protective gear, as well as rudimentary movement such as safely entering and exiting a patrol vehicle safely and quickly. Significant decreases in the ability to perform the overhead squat movement were also found (Table 4 and Figure 2B) with lower scores in the Gear condition in 23% of the population. The overhead squat is a full-body movement examining ankle, knee, hip, trunk, and shoulder kinetic patterns. The movement pattern became more difficult when officers donned the protective gear, and some officers were no longer able to complete the proper pattern in the gear condition. This decrease in full-body movement can impact police officers’ daily tasks and contribute to acute or chronic injury by altering biomechanical processes and patterns [7,10,11].

The sit-and-reach assessment also revealed the stark difference in physical ability for police officers in and out of the traditional protective gear (Figure 3C). This assessment is one of the most used flexibility tests for performance prediction and evaluation of police officers [31]. A 29% decrease in the lower back and hamstring flexibility and movement was found when officers completed the sit-and-reach assessment in the Gear condition. The alterations observed in the mobility and flexibility patterns of the officers may be primarily attributed to the additional weight, inherent rigidity, and inflexibility of the protective gear. These factors collectively impose constraints on the officers’ range of motion and impede their ability for functional movement. In comparison to the previous literature, the sample population of officers for this study tested lower overall on the sit-and-reach assessment. The mean of the sample population in the No Gear condition was recorded as nearly half of the mean distance in the previous literature [32]. Of the current population tested, 63% reported experiencing low back pain either on or off duty. It would be beneficial to investigate this stark difference in low back and hamstring flexibility as the decrease may be in conjunction with officers who are primarily on car patrol duties and sit in vehicles for the majority of their shifts.

This study revealed notable reductions in vertical jump proficiency among law enforcement officers upon donning their protective gear, indicating a decline in power output and output with the additional load imposed by the equipment. This decreased vertical jump height outcome was consistent with previous findings (Figure 3) [33]. The 16% decrease in power observed in the vertical jump assessment is of concern in police operations such as the pursuit or apprehension of suspects, scaling fences, negotiating obstacles, and navigating crowded environments during pursuits. The disparities in force production between the right and left lower extremities between conditions underscore the significance of understanding the implications of gear-induced alterations in power distribution and reliance. It is noteworthy that officers typically engage in strength and conditioning in athletic wear rather than protective gear with the goal of reducing stressors on the body. However, this may reduce adaptations needed to prepare officers for real-world operational demands.

The disparities in physical performance and power output between the Gear and No Gear conditions hold implications for law enforcement activities, potentially predisposing officers to a heightened risk of musculoskeletal injuries during operational duties while equipped with protective gear. A lack of distribution of leg dominance among the sample population hindered a comparison of force output changes based on leg dominance. Only 8.3% of the sample population stated left leg dominance. Therefore, we were unable to meet the threshold for properly powered testing to evaluate changes in power output between the left and right legs in the two gear conditions. It is common for individuals to have an imbalance of strength and power in comparison between the legs [34,35]. It was evident during the vertical jump assessment that most participants exhibited this similar imbalance through higher power output from one leg rather than an equal distribution of power output. An explanation of this imbalance may be attributed to past experiences such as injuries, sporting activities, and unintentional dominant training known as “lateral preference”. Examples of lateral preference may include always leading with one leg going up and down stairs, always kneeling on one knee, using the preferred leg more upon sit-to-stand movements, etc. [34,35]. Sporting events, particularly single-side sporting events such as tennis, baseball, and softball, and even performing layups in basketball lead to this lateral preference than can be seen later in life [35]. The aforementioned concerns and instances may contribute to the imbalance of power output observed in this population.

The ability to sustain balance is critical in the daily activities of law enforcement personnel. The heightened risk of falls or the potential for additional musculoskeletal injuries stemming from compromised balance and stability under the burden of added weight underscores the need to maintain optimal balance proficiency during law enforcement activities. Previous research has emphasized the connection between increased sway and sway velocity and its association with increased risks of slips, trips, and falls, especially with additional load carriage [27]. One study found that with loads of 9.1 kg, there was a 42% increase in participants hitting/tripping obstacles while attempting to step over them [36]. Previous research has found decreased functionality and adaptations in their investigation of postural sway and use of center-of-pressure measurements in both the general and tactical population [37,38]. Officers in the current study were able to maintain global balance and stability despite an increase in energy expenditure required to sustain adequate balance and stability, as demonstrated by the balance test.

As a part of this study, we assessed BMI and body composition to evaluate possible exacerbation of the reported issues with required police equipment. Body composition and excessive body fat percentage are becoming a more prominent concern in the general population and with tactical athletes [32]. The use of BMI as a predictive measure is a common standard in the general and tactical athlete populations [39,40,41]. BMI is a simple and cost-effective method to gain insight into the estimated overall health and body composition of an individual. However, research has shown that BMI is not the most accurate way of assessing body composition due to additional weight with increased muscle mass [42]. As individuals increase in muscle mass, overall weight is increased and may skew the numerical estimate of overall health from the BMI measure [43,44]. Thus, we assessed both measures in this study. Our analysis revealed differing interactions between body fat percentage and BMI in both the No Gear and Gear conditions. The vertical jump assessment indicated that individuals with higher BMI generated more overall maximum force than those with lower BMI in both conditions (Figure 4). In opposition, for jump height, relative maximum force, and maximum power in both the No Gear and Gear conditions, individuals with a lower body fat percentage performed better with higher, more powerful, and more relative force output than officers with higher body fat percentage. A similar finding was found with the modified pushup in which individuals with lower body fat performed better in both conditions. Although there was no statistical significance found in the sit-and-reach assessment when comparing BMI and body fat percentage interactions, the research team did not record negative numbers from the assessment. In the case that a participant could not reach the bracket of the sit-and-reach box, this was recorded as a “0”, which may have influenced these results. These findings illustrate that body fat percentage should be assessed rather than only BMI when associating body composition with performance measures. Additionally, more emphasis should be placed on maintaining a recommended body fat percentage rather than BMI alone for police officers to enhance performance and movements for daily occupational tasks.

Limitations of this study include an imbalance in biological sex representation within the police department in this study. The national biological sex ratio for officers is approximately 13.3% female and 86.7% male [45], whereas the current study only had 5.9% female officer participants. The sample size of our study poses an additional limitation. The participating police department was in a middle-sized town. However, 43.6% of officers participated, with a total of 72 police officers. Protective gear worn by the officers and the location of equipment placed on the duty belt were not standardized. We conducted this study with each officer’s personal gear and preference of location for all equipment to emulate actual field responses between the No Gear and Gear conditions.

## 5. Conclusions

These findings revealed important differences in mobility, flexibility, and power output when police officers are in athletic clothing compared to wearing full protective gear and daily uniform. Additionally, increased body fat percentage, as opposed to BMI, in police officers enhanced the limitations and decreased mobility, flexibility, and power output when the officers donned their protective gear. Although balance and stability did change while wearing protective gear, officers generally were able to maintain balance despite increased energy expenditure. The results of the current study highlight the need for continued evaluation of alternative means for protective equipment for daily police officer use. These alternatives should emphasize reducing impacts on physical abilities such as movement and power while maintaining the indispensable safety that is needed from mandatory protective equipment. Our work also suggests that body composition analysis should be used for evaluations instead of BMI whenever possible. It is critical for those who protect and serve to have optimal protective equipment to complete daily requirements while maintaining their own safety.

## Figures and Tables

**Figure 1 jfmk-10-00344-f001:**
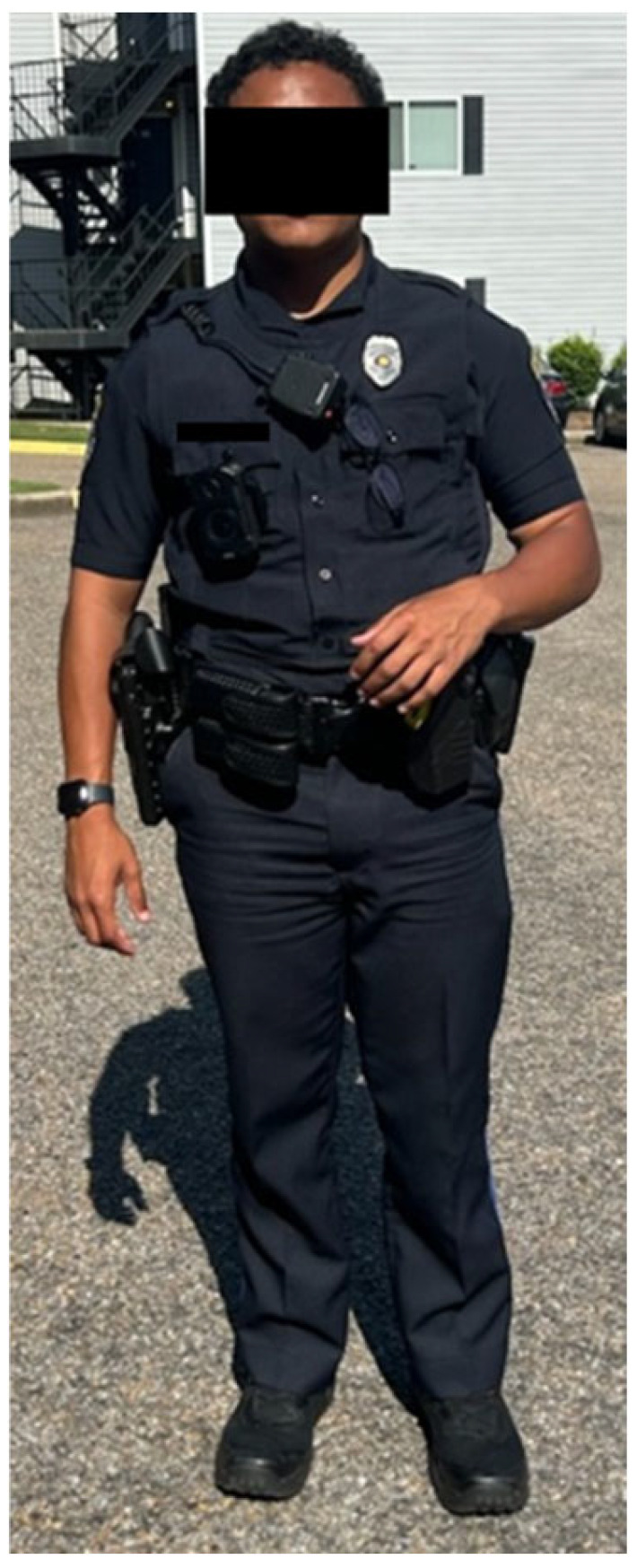
Police officer in protective gear and duty uniform.

**Figure 2 jfmk-10-00344-f002:**
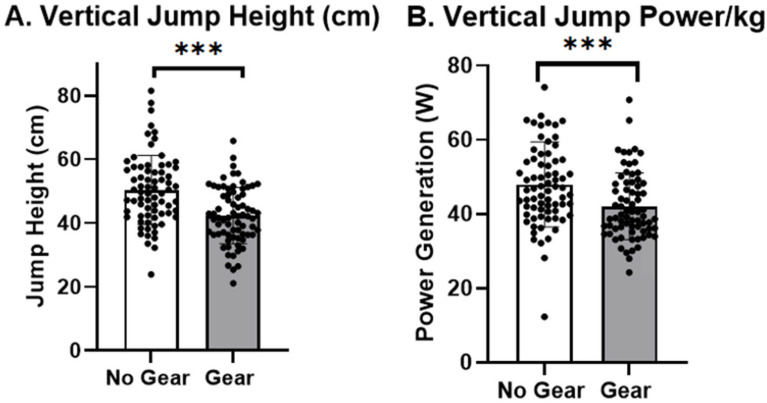
Vertical jump measures between Gear and No Gear conditions. Cm = centimeters; W = watts; *** *p* <0.001.

**Figure 3 jfmk-10-00344-f003:**
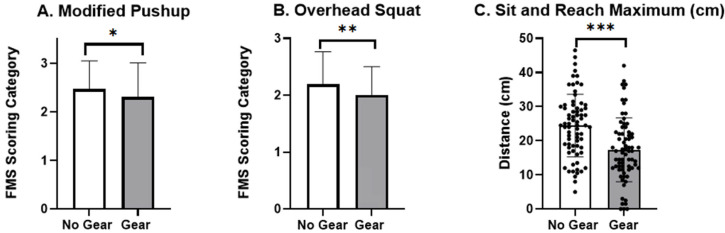
Comparison mobility and flexibility assessments in the Gear and No Gear conditions. Note: Score of 1: Inability to perform or complete the functional movement pattern. Score of 2: Ability to perform the functional movement pattern, but some degree of compensation was noted. Score of 3: Unquestioned ability to perform the functional movement pattern. Cm = centimeters; * = statistical significance; * *p* = 0.05–0.02; ** *p* = 0.01–0.002; *** *p* < 0.001.

**Figure 4 jfmk-10-00344-f004:**
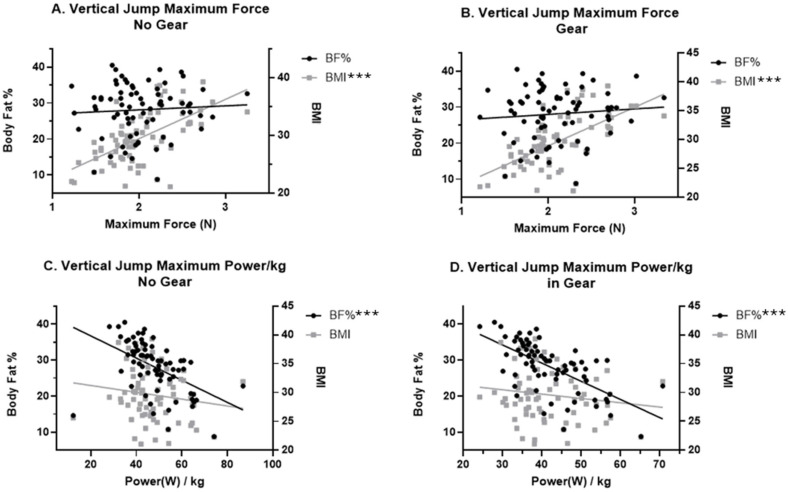
Vertical jump outcomes and body composition in No Gear and Gear conditions. *** *p* < 0.001.

**Table 1 jfmk-10-00344-t001:** Standard weight of police gear and equipment.

Gear/Equipment	Weight (lbs)	Weight (kg)
Protective Vest	6.4	2.9
Weapon (Pistol)	2.4	1.1
Taser	1.0	0.5
Handcuffs	0.4	0.2
Radio	1.8	0.8
Body Camera	0.4	0.2
Loaded Magazine	0.6	0.3
Belt (Unloaded)	3.4	1.5
Pepper Spray	0.3	0.1
Flashlight	0.4	0.2

Note: lbs = pounds; kg = kilograms.

**Table 2 jfmk-10-00344-t002:** Participant demographics.

Participants	72		
Male/Female	68/4		
Right/Left Hand Dominance	R-65		L-6
Weapon Hand Dominance	R-67		L-4
Right/Left Foot Dominance	R-65		L-6
Age (yrs)	36.9	±	9.7
Height (cm)	178.6	±	7.5
Body Mass (kg)	95.1	±	11.8
Body Fat Percentage	28.1	±	7.1

Notes: Values are reported as mean ± SD; yrs = years; cm = centimeters; kg = kilograms; R = right; L = left.

**Table 3 jfmk-10-00344-t003:** Balance: No Gear vs. Gear (path length mean velocity).

Predictor	B (SE)	T	*p*-Value
Intercept (No Gear Condition)	5.19 (0.29)	17.62	*p* < 0.001
Condition (Gear vs. No Gear)	−0.60 (0.42)	−1.43	*p* = 0.15

Note: R^2^ = 0.0014; F_(1,142)_ = 2.05.

**Table 4 jfmk-10-00344-t004:** Functional movement assessments in Gear and No Gear conditions.

Assessment	No Gear N%		Gear N%		% Change
Modified Pushup					
1	3	4.17%	10	13.89%	↑9.72%
2	32	44.44%	30	41.67%	↓2.77%
3	37	51.39%	32	44.44%	↓6.95%
Overhead Squat					
1	6	8.33%	9	12.5%	↑4.17%
2	46	63.89%	54	75%	↑11.11%
3	20	27.78%	9	12.5%	↓15.28%
Sit-and-Reach					Distance Change
Distance (cm)	24.45 cm		17.37 cm		↓7.08 cm

Note: Score of 1: Inability to perform or complete the functional movement pattern. Score of 2: Ability to perform the functional movement pattern, but some degree of compensation was noted. Score of 3: Unquestioned ability to perform a functional movement pattern. cm is centimeters. ↑ indicates an increase in change, ↓ indicates a decrease in change.

## Data Availability

The data presented in this study are available on request from the corresponding author due to the nature of the participant data of currently working police officers.

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
