# Peer review of "Protective Gear Negatively Impacts Police Officer Mobility, Stability, and Power Generation"

_jfmk, 2025, doi:10.3390/jfmk10030344_

Round 1

Reviewer 1 Report

Comments and Suggestions for Authors

All statements need a reference, please add reference in the following lines: 27-31-62-64-66

The material and methods section is very messy, i suggest order with subsections (2.1 sample, 2.2 procedures, 2.3 body composition, and one for each physical test and statistical analysis)

Statistical analysis

Your stastistical analysis is very confuse, MANOVA analysis is wrong you should be perform ANCOVA analysis where Body fat and body mass are covariables or correlations between performance test and BF and BM, with and without gear for separate.

add measures of effects size (cohen's d for t test and partial eta squared for ANOVA and cohen's d for post-hoc comparisons)

Results

please improved formatting of tables

you have figure wiht bar and jitter plot and only with bar, please add the jitter in all figures.

Add conclusion section

Author Response

Thank you for your time and contribution to improving our manuscript. We have addressed each concern and have inserted the changes into the re-submitted manuscript. Please see our comments below.

Reviewer 1

All statements need a reference, please add reference in the following lines: 27-31-62-64-66

Thank you for the notification. We have increased the amount of references in the listed areas to ensure more robust citations including:

 Line:

 27 :

1.Larsen, L.B.; Ramstrand, N.; Tranberg, R. Duty belt or load-bearing vest? Discomfort and pressure distribution for police driving standard fleet vehicles. Appl Ergon 2019, 80, 146-151, doi:10.1016/j.apergo.2019.05.017.

2.Lenton, G.K.; Saxby, D.J.; Lloyd, D.G.; Billing, D.; Higgs, J.; Doyle, T.L.A. Primarily hip-borne load carriage does not alter biomechanical risk factors for overuse injuries in soldiers. J Sci Med Sport 2019, 22, 158-163, doi:10.1016/j.jsams.2018.06.013.

31:

1.Larsen, L.B.; Ramstrand, N.; Tranberg, R. Duty belt or load-bearing vest? Discomfort and pressure distribution for police driving standard fleet vehicles. Appl Ergon 2019, 80, 146-151, doi:10.1016/j.apergo.2019.05.017.

3.Schram, B.; Hinton, B.; Orr, R.; Pope, R.; Norris, G. The perceived effects and comfort of various body armour systems on police officers while performing occupational tasks. Annals of occupational and environmental medicine 2018, 30, 1-10.

62:

  1. Cheuvront, S.N.; Goodman, D.A.; Kenefick, R.W.; Montain, S.J.; Sawka, M.N. Impact of a protective vest and spacer garment on exercise-heat strain. European journal of applied physiology 2008, 102, 577-583.
  2. Hasselquist, L.; Bensel, C.K.; Brown, M.L.; O’Donovan, M.P.; Coyne, M.; Gregorczyk, K.N.; Kirk, J. Physiological, biomechanical, and maximal performance evaluation of medium rucksack prototypes. DTIC Doc 2013.
  3. Larsen, B.; Netto, K.; Skovli, D.; Vincs, K.; Vu, S.; Aisbett, B. Body armor, performance, and physiology during repeated high-intensity work tasks. Military medicine 2012, 177, 1308-1315.

64:

  1. Dempsey, P.C.; Handcock, P.J.; Rehrer, N.J. Impact of police body armour and equipment on mobility. Appl Ergon 2013, 44, 957-961, doi:10.1016/j.apergo.2013.02.011.
  2. Shim, A.; Shannon, D.; Waller, M.; Townsend, R.; Obembe, A.; Dial, M. Tactical vests worn by law enforcement: is this improving stability for optimal job performance? International journal of occupational safety and ergonomics 2023, 29, 177-180

66:

  1. Lewinski, W.J.; Dysterheft, J.L.; Dicks, N.D.; Pettitt, R.W. The influence of officer equipment and protection on short sprinting performance. Appl Ergon 2015, 47, 65-71, doi:10.1016/j.apergo.2014.08.017.
  2. Lockie, R.; Dawes, J.J.; Sakura, T.; Schram, B.; Orr, R.M. Relationships between physical fitness assessment measures and a workplace task-specific physical assessment among police officers: A retrospective cohort study. The Journal of Strength & Conditioning Research 2023, 37, 678-683.

The material and methods section is very messy, i suggest order with subsections (2.1 sample, 2.2 procedures, 2.3 body composition, and one for each physical test and statistical analysis)

We appreciate your assistance in ensuring a clean manuscript and have added in these additional subsections for ease of reading and clarity.

Statistical analysis

Your stastistical analysis is very confuse, MANOVA analysis is wrong you should be perform ANCOVA analysis where Body fat and body mass are covariables or correlations between performance test and BF and BM, with and without gear for separate.

Upon additional investigation we agree the statistical analysis section was not written for proper clarity and specificity. We have re-written and clarified proper statistical methodology for each of the sections.

add measures of effects size (cohen's d for t test and partial eta squared for ANOVA and cohen's d for post-hoc comparisons)

We have added in the additional statistical information including the above mentioned areas.

Results

please improved formatting of tables

you have figure wiht bar and jitter plot and only with bar, please add the jitter in all figures.

Thank you for your remarks. Initially we did not add in the jitter plot with the ordinal measures as they tend to look compacted and hinder the comprehension of the figures. We have re-added the jitter plots onto these figures.

Add conclusion section

Conclusion section has been added and more properly clarified. 

Reviewer 2 Report

Comments and Suggestions for Authors

This study examined the impact of wearing uniform and protective gear, with an average weight of approximately 9 kg, on balance, functional movements, and lower limb power generation ability in a group of police officers. All measurements were obtained using standardised procedures and high-quality equipment, ensuring the accuracy of the reported results. While the study offers limited scientific novelty, its findings are relevant for understanding the physical constraints imposed on police officers during duty and the potential injury risks associated with mandatory uniform and protective equipment. Nevertheless, in my opinion several important issues need to be addressed before the manuscript can be considered for publication. The major concern lies in the statistical methodology, which is not clearly described and, in some instances, appears to be inappropriate.

Materials and methods

Lines 101-105. Please, provide a detailed description of the method used to derive the power output for each lower limb during the vertical jumps.

Line 110. The sentence “…scores on a scale of zero to one…” suggests a continuous range between 0 and 1. I recommend avoiding such ambiguity by reporting that FMS pain provocation tests were scored 0 or 1.

Lines 125-136 (Statistical methods). T-tests are generally not appropriate for comparing conditions when the variables are measured on a 3-point ordinal scale, as is the case for many variables in the study. I suggest re-analysing these comparisons using a non-parametric test, such as the Wilcoxon signed-rank test to account for the ordinal nature of the data and the paired design.

Lines 130-132. The methods section reports that MANOVA was used to evaluate the “interaction and influence” of body fat (%BF) and body mass index (BMI) on several outcome variables. Yet, %BF and BMI are continuous variables therefore not appropriate for MANOVA unless previously categorised into discrete groups, a step not described in the manuscript. If %BF and BMI were used as continuous predictors the appropriate approach would likely be a MANCOVA rather than MANOVA. This point needs to be substantially clarified, including how %BF and BMI were handled and what statistical test was actually applied.

The study reports a considerable number of pairwise comparisons to assess the impact of gear on several variables. However, except for ANOVA testing, no correction for multiple comparisons is described. Please indicate whether adjustments (e.g. Bonferroni correction) were applied to control for inflated type I error rate. If no correction was performed, the analysis should be repeated with adjustment for multiple testing.  

Results

Lines 166-176. This paragraph and the data presented in Table 5 are difficult to understand within a context of a MANOVA analysis. The relationship the MANOVA analyses, the reported statistics in Table 5 is not clearly explained. This section should be revised and clarified, in line with my earlier comment regarding the suitability and implementation of MANOVA in this study.

Lines 166-167 and Fig. 2. I question the relevance of comparing mass in the Gear and No Gear conditions, as the increase in mass is solely due to the weight of the uniform and equipment. Also, it is not strictly accurate to refer to this measure as “body mass” when weighing officers with the gear on, since the added weight does not represent the body weight.

Discussion

Figs. 3, 4, 5 and 6 are not reported or described in the results section but appear only in the Discussion. The results shown in these figures should be presented and explained in the Results section. Also, I recommend avoiding the placement of statistical significance symbols (***) next to the variable names on the y-axis, as it happens in Figs 5C, 5D, 6A and 6B. These symbols should instead be placed closed to the relevant data.

Author Response

Thank you for your time and contribution to improving our manuscript. We have addressed each concern and have inserted the changes into the re-submitted manuscript. Please see our comments below.

Reviewer 2

This study examined the impact of wearing uniform and protective gear, with an average weight of approximately 9 kg, on balance, functional movements, and lower limb power generation ability in a group of police officers. All measurements were obtained using standardised procedures and high-quality equipment, ensuring the accuracy of the reported results. While the study offers limited scientific novelty, its findings are relevant for understanding the physical constraints imposed on police officers during duty and the potential injury risks associated with mandatory uniform and protective equipment. Nevertheless, in my opinion several important issues need to be addressed before the manuscript can be considered for publication. The major concern lies in the statistical methodology, which is not clearly described and, in some instances, appears to be inappropriate.

Materials and methods

Lines 101-105. Please, provide a detailed description of the method used to derive the power output for each lower limb during the vertical jumps.

A more thorough description of the Leonardo Mechanograph and measurement recording has been added to this section.

Line 110. The sentence “…scores on a scale of zero to one…” suggests a continuous range between 0 and 1. I recommend avoiding such ambiguity by reporting that FMS pain provocation tests were scored 0 or 1.

Thank you for assisting with this clarification and typo. We have changed this to “or”.

Lines 125-136 (Statistical methods). T-tests are generally not appropriate for comparing conditions when the variables are measured on a 3-point ordinal scale, as is the case for many variables in the study. I suggest re-analysing these comparisons using a non-parametric test, such as the Wilcoxon signed-rank test to account for the ordinal nature of the data and the paired design.

We agree and thank you for your attention to this oversight. Upon additional investigation we agree the statistical analysis section was not written for proper clarity and specificity. We have re-written and clarified proper statistical methodology for each of the sections.

Lines 130-132. The methods section reports that MANOVA was used to evaluate the “interaction and influence” of body fat (%BF) and body mass index (BMI) on several outcome variables. Yet, %BF and BMI are continuous variables therefore not appropriate for MANOVA unless previously categorised into discrete groups, a step not described in the manuscript. If %BF and BMI were used as continuous predictors the appropriate approach would likely be a MANCOVA rather than MANOVA. This point needs to be substantially clarified, including how %BF and BMI were handled and what statistical test was actually applied.

Thank you for bringing this to our attention. We agree. As stated above we have addressed this by ensuring our statistical analysis and more clearly writing the statistical procedure for each section.

The study reports a considerable number of pairwise comparisons to assess the impact of gear on several variables. However, except for ANOVA testing, no correction for multiple comparisons is described. Please indicate whether adjustments (e.g. Bonferroni correction) were applied to control for inflated type I error rate. If no correction was performed, the analysis should be repeated with adjustment for multiple testing.  

We have addressed this concern as described above.

Results

Lines 166-176. This paragraph and the data presented in Table 5 are difficult to understand within a context of a MANOVA analysis. The relationship the MANOVA analyses, the reported statistics in Table 5 is not clearly explained. This section should be revised and clarified, in line with my earlier comment regarding the suitability and implementation of MANOVA in this study.

Lines 166-167 and Fig. 2. I question the relevance of comparing mass in the Gear and No Gear conditions, as the increase in mass is solely due to the weight of the uniform and equipment. Also, it is not strictly accurate to refer to this measure as “body mass” when weighing officers with the gear on, since the added weight does not represent the body weight.

We agree and have removed Figure 2 and have renumbered the figures and references.

Discussion

Figs. 3, 4, 5 and 6 are not reported or described in the results section but appear only in the Discussion. The results shown in these figures should be presented and explained in the Results section. Also, I recommend avoiding the placement of statistical significance symbols (***) next to the variable names on the y-axis, as it happens in Figs 5C, 5D, 6A and 6B. These symbols should instead be placed closed to the relevant data.

Thank you, we agree with your view on the figures and references of the figures. We have moved the figures and additional references have been added throughout the manuscript and results section for the figures and we have removed the significance symbols from the y-axis.